# Dietary Approaches to Stop Hypertension via Indo-Mediterranean Foods, May Be Superior to DASH Diet Intervention

**DOI:** 10.3390/nu15010046

**Published:** 2022-12-22

**Authors:** Ram B. Singh, Fatemeh Nabavizadeh, Jan Fedacko, Dominik Pella, Natalia Vanova, Patrik Jakabcin, Ghizal Fatima, Rie Horuichi, Toru Takahashi, Viliam Mojto, Lekh Juneja, Shaw Watanabe, Andrea Jakabcinova

**Affiliations:** 1Halberg Hospital and Research Institute, Moradabad 244001, India; 2Department of Cardiology, Emirates Hospital, Dubai 999041, United Arab Emirates; 3Department of Gerontology and Geriatric, PJ Safarik University and MEDIPARK—University Research Park, PJ Safarik University, 1, 041-90 Kosice, Slovakia; 41st Department of Cardiology, PJ Safarik University Faculty of Medicine and East Slovak Institute for Cardiovascular Disease, 040-11 Kosice, Slovakia; 5Department of Internal Medicine, PJ Safarik University and Agel Hospital Kosice-Saca, 040-11 Kosice, Slovakia; 6Department of Social and Clinical Pharmacy, Faculty of Pharmacy in Hradec Králové, Charles University, 10000 Prague, Czech Republic; 7Era Medical College, Era University, Lucknow 226001, India; 8Department of Food Sciences and Nutrition, Faculty of Human Environmental Sciences, Mukogawa Women’s University, Nishinomiya City 663-8558, Japan; 9Department of Nutrition, Faculty of Nutrition, Kanazawa Gakuin University, Kanazawa City 920-1392, Japan; 10Department of Internal Medicine, Comenius University, 813-72 Bratislava, Slovakia; 11Executive Vice President, Kameda Seika Co., Ltd., Tokyo 160-0005, Japan; 12Life Science Association, Tokyo 160-0005, Japan; 13Department of Gerontology and Geriatric, Faculty of Medicine, PJ Safarik University and MEDIPARK—University Research Park, PJ Safarik University, 040-11 Kosice, Slovakia

**Keywords:** high blood pressure, Mediterranean diet, anti-inflammatory foods, western diet

## Abstract

Western-type diet with high salt and sugar, sedentary behavior, obesity, tobacco and alcoholism are important risk factors for hypertension. This review aims to highlight the role of western diet-induced oxidative stress and inflammation in the pathogenesis of hypertension and the role of various types of diets in its prevention with reference to dietary approaches to stop hypertension (DASH) diet. It seems that it is crucial to alter the western type of diet because such diets can also predispose all CVDs. Western diet-induced oxidative stress is characterized by excessive production of reactive oxygen species (ROS) with an altered oxidation-reduction (redox) state, leading to a marked increase in inflammation and vascular dysfunction. Apart from genetic and environmental factors, one important cause for differences in the prevalence of hypertension in various countries may be diet quality, deficiency in functional foods, and salt consumption. The role of the DASH diet has been established. However, there are gaps in knowledge about the role of some Indo-Mediterranean foods and Japanese foods, which have been found to decrease blood pressure (BP) by improving vascular function. The notable Indo-Mediterranean foods are pulses, porridge, spices, and millets; fruits such as guava and blackberry and vegetables, which may also decrease BPs. The Japanese diet consists of soya tofu, whole rice, in particular medical rice, vegetables and plenty of fish rich in fish oil, fish peptides and taurine that are known to decrease BPs. Epidemiological studies and randomized, controlled trials have demonstrated the role of these diets in the prevention of hypertension and metabolic diseases. Such evidence is still meager from Japan, although the prevalence of hypertension is lower (15–21%) compared to other developed countries, which may be due to the high quality of the Japanese diet. Interestingly, some foods, such as berries, guava, pumpkin seeds, carrots, soya beans, and spices, have been found to cause a decrease in BPs. Omega-3 fatty acids, fish peptide, taurine, dietary vitamin D, vitamin C, potassium, magnesium, flavonoids, nitrate and l-arginine are potential nutrients that can also decrease BPs. Larger cohort studies and controlled trials are necessary to confirm our views.

## 1. Introduction

Hypertension has become a public health problem, and there are approximately 1.31 billion patients with hypertension, as per WHO estimates [1,2]. In the USA and other developed countries, as well as Mediterranean countries such as Greece and Spain, the prevalence of hypertension (defined as those receiving drug therapy or having a systolic pressure of ≥140 mmHg and/or a diastolic pressure ≥90 mmHg) was approximately 30% which has remained relatively constant since 1999 [3,4,5]. The prevalence of hypertension after adjustment of age in Japan is approximately 21% [6], which is lower than in other developed countries, Mediterranean countries and the prevalence rate in India (25%) and other developing countries [1,2,5]. Hypertension is a serious chronic disease known to enhance the risk of diseases of the brain, kidney, heart, and other organs and is a major cause of premature death worldwide [2]. The hypertension burden is disproportionately high in developing countries, where more than 66% of patients are present [1,2]. The cause may be the rapid increase in risk factors as a result of the nutritional transition from poverty to affluence in recent decades [1,2].

Risk factors for non-communicable diseases (NCDs), including hypertension, often begin early in life and continue through adulthood. There is evidence that both prevention and treatment are necessary for reversing the epidemic of hypertension. “The WHO Global NCD action plan is a road map with policy options implemented from 2013 to 2020 focusing on the four shared, modifiable behavioral risk factors that are linked to the four preventable NCDs,” including hypertension [1,2]. Of these risk factors, modification of the western type of diet is crucial, because such diets can also predispose obesity, hyperlipidemia, diabetes, coronary artery diseases (CAD) and stroke. One important cause for differences in the rate of hypertension prevalence in various countries may be due to diet quality, in particular salt, vegetable, fruit and fish consumption [1,2,3,7]. In the United States, dietary approaches to stop hypertension (DASH) diet **was** designed for the management of hypertension due to its high potassium and lower sodium contents [7,8,9]. It seems that there are a few gaps in our understanding and knowledge about the role of a western type of diet-induced oxidative stress and inflammation, causing an increased risk of hypertension. This review is relevant because a comparison of the DASH diet with a Mediterranean diet in patients with various glucose regulation states revealed that adherence to the Mediterranean diet was associated with a significant **greater** decline in all-cause mortality in a general population [3]. The exact mechanisms, of how Mediterranean foods, in particular Indo-Mediterranean foods as well as Japanese foods, reduce BPs by improved vascular function need clarification. Because certain foods and nutrients present in the concerned diets, may have potential anti-inflammatory and antihypertensive activity as a determinant of diet quality. This selected review aims to highlight the role of DASH diets, as compared with Indo-Mediterranean, Japanese and Mediterranean types of diets, in the management of hypertension. We used the DASH diet, Japanese diet, Mediterranean diet, fruits, vegetables and nutrient-rich diet in hypertension as keywords for search, then selected the concerned studies for review.

## 2. Effects of Dietary Approaches to Stop Hypertension Diet

In earlier studies DASH diet was found to cause a significant decline in blood pressure (BPs), despite optimal salt intake [7,8,9]. Since the DASH diet was developed for Americans, it is difficult for Asian populations to adapt to the same DASH diet because of the preferences in taste and flavor of these subjects [10,11,12]. There may be difficulty in designing a menu that matches the content of nutrients of the DASH diet by using Asian dishes. Since it has been difficult for Japanese people to consume the DASH diet on a regular basis, a Japanese DASH diet has been designed to reduce BP among untreated subjects with mild hypertension [11]. The Japanese DASH diet is also rich in fruits and vegetables, with low-fat dairy products and provides 8.0 g salt per day with a two-fold increase in potassium, magnesium, calcium, flavonoids and fiber, which are important components in the average Asian diet [11,12]. The value of the DASH diet is now established and accepted by NIH [13] as well as by other centers [14,15,16,17] apart from Japan [11,12]. In earlier studies, an Indo-Mediterranean diet or a traditional Mediterranean style diet comprised of traditional foods, fruits, vegetables, pulses and legumes, has been used for decreasing BPs [18,19,20] as well as the risk of morbidity and mortality due to CVDs [21,22,23]. A revised Scientific Statement of the International College of Cardiology and International College of Nutrition-2011, for the prevention of CVDs and diabetes [24] and recently, a review on why and how the Indo-Mediterranean diet may be superior to the DASH diet has also been published [25].

The DASH diet is a dietary pattern promoted by the U.S.-based National Heart, Lung, and Blood Institute to prevent and control hypertension. This trial enrolled 459 adult patients with systolic blood pressure (BP) of lower than 160 mm Hg and diastolic BP of 80 to 95 mm Hg, who were asked to eat the usual American diet for 3 weeks before entry to study [7]. The trial found that apart from decrease in BPs, there was a significant decline in BP variability. The DASH diet is rich in fruits, such as berries, apples, oranges, bananas, and apricots; vegetables, such as green leaves and green beans; whole grains, such as oats and beans; low-fat dairy foods, such as milk, yogurt, cheese, and vegetable oils, as well as low red meat, egg and fish [7,13]. It has been reported that the DASH diet has been declared “Best Diets for Healthy Eating” and “Best Heart Healthy Diets” in the 2021 [13]. DASH diet is comprised of meals that are dense in nutrient-around low-fat dairy products, whole grains, fruits and vegetables, as well as being low in red meat and preserved meat and sugar. This diet also includes poultry, fish, nuts, beans, and healthy vegetable oils and ranks second for “Best Diet Overall”.

Appel and coworkers proposed the DASH diet, approximately 20 years before with a research grant from the National Heart, Lung, and Blood Institute (NHLBI) in the United States. In the DASH diet trial, at baseline, the mean systolic and diastolic BPs were 131.3 ± 10.8 mm Hg and 84.7 ± 4.7 mm Hg, respectively [7]. The combination diet reduced systolic and diastolic BPs by 5.5 and 3.0 mm Hg more, respectively, compared to the control diet (*p* < 0.001 for each); the fruits-and-vegetables diet reduced systolic BP by 2.8 mm Hg more (*p* < 0.001) and diastolic BP by 1.1 mm Hg more than the control diet (*p* = 0.07). For all subjects with hypertension (n = 133) having systolic BP, > or =140 mm Hg; diastolic BP, > or =90 mm Hg; or both, the DASH diet reduced systolic and diastolic BPs by 11.4 and 5.5 mm Hg more, respectively, than the control diet. Subjects without hypertension (n = 326) showed that the corresponding reductions in BPs were 3.5 mm Hg (*p* < 0.001) and 2.1 mm Hg (*p* = 0.003).

Several trials have reported that the plan of eating is helpful in reducing BPs and blood cholesterol, which are predisposing factors for coronary disease [7,8,9,13,14]. In adults, those subjects that consume the DASH diet, without reducing salt intake can decrease their BP within a period of a few weeks, compared to the standard diet group [9]. However, subjects who decreased their intake of sodium from 3450 mg to 2300 mg or lower daily intake showed even more decline in BPs. Those patients who preferred lower sodium content in the DASH diet, equal to half of the sodium intake, found the greatest benefit. Apart from the reduction in salt intake, the low-sodium-DASH, an additional alteration includes a reduction of 10% of per day carbohydrates for unsaturated fats or proteins. Most interestingly, the meals contain foods that are rich in potassium, magnesium, calcium, flavonoids, protein and fiber, while unhealthy nutrients such as sugar, trans fat, sodium, sugar, and saturated fats are low [13]. Other health advice also integrate regular physical activity, expecting a healthy body weight, and limiting or cessation of alcohol. Figure 1.

In a meta-analysis published in 2021, comprised of a total of 10 randomized, controlled trials, treatment with the modified DASH diet was associated with a decline in mean systolic BP (−3.26 mmHg; *p* = 0.006) and diastolic (−2.07 mmHg; *p* = 0.01), compared with control diet [16]. The modified DASH diet decreased systolic BP to a higher level in trials with a mean baseline BP ≥ 140/90 mmHg compared with <140/90 mmHg, as compared with a standard diet. A decline in diastolic BP was greater when the mean body mass index was ≥30 kg/m^2^ than <30 kg/m^2^. The decline in diastolic BP was more significant in studies with a follow-up time of >8 weeks compared with ≤8 weeks. Mean waist circumference (difference: 1.57 cm; and triglyceride concentration (difference: 1.04 mol/L; also revealed a significant decline in the DASH diet group. It is clear that the modified DASH diet may cause a significant decline in BPs, waist circumference, and triglyceride levels in patients with hypertension.

In view of significant methodological and clinical differences in trials, another meta-analysis included subjects with and without hypertension, for a comprehensive assessment of the effects of the DASH diet on BP in adults [26], with due consideration for confounders. Among 30 clinical trials (n = 5545 participants), included, the BP-lowering effects of the DASH diet were compared with a control diet in hypertensive and non-hypertensive adults. Compared with a control diet, the DASH diet decreased both systolic BP and diastolic BP by mean differences: −3.2 mm Hg; 95%; *p* < 0.001, and −2.5 mm Hg; *p* < 0.001, respectively. The DASH diet, compared with the standard diet, also decreased systolic BP levels to a higher extent in trials with sodium intake >2400 mg/d compared to trials with sodium intake ≤2400 mg/d, whereas both SBP and DBP were reduced more in trials with mean age <50 y than in trials of elderly subjects [26]. Treatment with the DASH diet was associated with a significant decline in BP in adults with and without high BP. The intake of higher per day sodium and younger age caused a greater decline in BP.

## 3. Effects of Mediterranean Type of Diet on Hypertension

The effects of Mediterranean style diets appear to be consistent with the DASH diet in reducing BPs and blood lipids. In an earlier meta-analysis, as compared to the standard diet and all other active intervention diets, the Mediterranean style diet caused a significant decline in BPs, with mean differences: −1.5 mm Hg; *p* = 0.035, systolic and −0.9 mm Hg, −0.3; *p* = 0.002, diastolic, respectively [27]. Interestingly, the comparison of only the usual diet with the Mediterranean diet found a decline in both systolic and diastolic BPs. The comparison of all other active intervention diets or only the low-fat diet with the Mediterranean style diet did not show a decline in systolic and diastolic BP. The Mediterranean style diet caused a greater decline in diastolic BP levels in trials with mean baseline SBP ≥130 mm Hg, while both systolic BP and diastolic BP were decreased greater in trials with a mean follow-up period ≥16 weeks [27]. It is clear that regular consumption of s Mediterranean style diet may be causative for a relatively small yet significant decline in BP. Higher baseline levels of systolic BP and longer duration of follow-up can enhance the lowering in the BP due to the intervention diet. The quality of evidence was rated as moderate for both outcomes according to the grading of recommendations, assessment, development and evaluation (GRADE) approach. Another meta-analysis, including 54 observational studies, found that systolic BP was reduced in the subgroup with greater adherence to the Mediterranean style diet group: −0.08, whereas no differences were found in the diastolic BP between the high and low adherence to Mediterranean diet groups: systolic: −0.07, 0.00 [28]. The mean diastolic BP of all the studies included for various adherence groups was within desirable limits (<90 mmHg). It is clear that greater adherence to a Mediterranean style diet may have a positive influence on systolic BP without any significant effects on diastolic BP. However, further randomized, controlled trials would be needed due to the variable definitions of lesser and greater adherence and because the results were based on observational studies.

Another meta-analysis, including 19 randomized, controlled trials, reported details of 4137 subjects and 16 cohort studies, including data from 59,001 subjects [29]. Intervention with the Mediterranean style diet was associated with a significant decline in systolic BP and diastolic BP with a mean decrease of −1.4 mmHg, *p* = 0.007, *I*^2^ = 53.5%, and −1.5 mmHg, *p* = 0.013, *I*^2^ = 71.5%, versus control, respectively. The longer duration of the study and greater systolic BP at baseline was associated with an increased decline in BP in response to a Mediterranean style diet (*p* < 0.05). In cohort studies, the odds of having high BP were 13% lesser with greater versus lesser Mediterranean style diet adherence (*p* = 0.017, *I*^2^ = 69.6%) [29]. It is clear that a Mediterranean style diet with a higher score is an effective nutritional method to reduce BP, in particular when the baseline BP is also higher and may cause decreased risk of CVDs.

## 4. Effects of Indo-Mediterranean Style Diet on Hypertension

Recently, it has been emphasized that the Indo-Mediterranean diet is more effective in decreasing CVDs and diabetes due to its high food diversity and nutrient density [25]. Epidemiological studies revealed that an unhealthy diet rich in energy is a risk factor for hypertension [30], whereas increased intake of fruits and vegetables may be inversely associated with the risk of hypertension [31]. In 1990, much before publication of the DASH diet, Singh and co-workers found out the role of fruits; guava, blackberry, apples, vegetables; green leafy, gourds, onion, garlic and legumes; green beans, black beans, kidney beans along with mustard oil in decreasing BPs [32]. In a controlled trial, 197 patients aged 25–65 with hypertension receiving diuretics were randomized to either an intervention diet (group A, 97 cases) or a control diet (group B, 100 cases). The intervention diet included fruits, vegetables and legumes, as described above, with a greater content of potassium (K), magnesium (Mg) and polyunsaturated fat, and fiber compared to the diet in the control group. Table 1.

After a follow-up of one year, patients with resistant hypertension were significantly fewer in group A compared to group B (n = 5 vs. 17, *p* < 0.01), respectively. Mean systolic (148.22 ± 10.1 vs. 160 ± 12.0 mm Hg) and diastolic (90.2 ± 4.84 vs. 103.3 ± 5.8 mm Hg) BPs in the intervention group A were significantly decreased compared to mean BPs, in group B, respectively and compared to baseline mean BPs (systolic 152.2 ± 12.8 mm Hg) and (diastolic 99.8 ± 7.2 mm Hg). The levels of mean serum Mg (1.86 ± 0.22 mg/dL) and K (4.86 ± 0.39 mg/dL) in the experimental group were significantly greater compared to mean levels of 1.56 ± 0.11 and 4.0 ± 0.29 mg/dL, respectively, in the control group. The increase in Mg and K levels in the treatment group may be due to increased intake of fruits, vegetables and legumes, which were significantly greater in the treatment group than in control subjects [32]. It is clear that a diet with a lower Na/K ratio and rich in fiber and flavonoids, K and Mg, and polyunsaturated fatty acids may cause a substantial decline in BPs.

It seems that loss of weight through a diet low in energy, with high-polyunsaturated fatty acids, fiber and potassium, can independently reduce BPs in hypertension. In another randomized trial, 416 patients with hypertension were administered either a low-energy Indo-Mediterranean type of diet (n = 106), a low-energy control diet (n = 104), an optimal-energy protective diet (n = 104), or an optimal-energy, control pre-experimental diet (n = 102) in conjunction with drug therapy in a single-blind trial [33]. Both intervention groups took significantly (*p* < 0.02) lesser energy per day than both control Groups. Both the treatment groups also got significantly (*p* < 0.02) greater energy as per the quantity of polyunsaturated fatty acids, complex carbohydrates, K, and Mg in the form of vegetables, fruits, and legumes, compared to the control groups. After a follow-up of 90 days, a significant decline in blood lipids occurred in the prudent diet group compared with the control diet group. Patients in the low-caloric treatment group revealed a weight loss of approximately 10 kg, without any changes in weight in the control diet group. Interestingly, at the end of the study, the weight loss showed a significant decline in BPs in both treatment groups (22/18 mmHg) and (16/13 mmHg), respectively, compared with control subjects. There was a significant decrease in BPs also in partial intervention group C (13/10 mmHg), compared with control patients, while drug therapy, salt consumption and physical activity scores revealed no differences in all the four groups.

In a previous randomized trial, 458 high-risk subjects aged 25–63, including patients with hypertension, were advised to either take a diet rich in fruits, vegetables and legumes and vegetable oil (mustard or peanut oil) (group A, n = 228) or a standard diet (group B, n = 230) in a randomized, single-blind fashion [34]. Group A received a significantly greater amount (>400 g/day) of fruits, vegetables and legumes in conjunction with vegetable oils (vegetable proteins, complex carbohydrates, polyunsaturated fats, fiber, K, Mg, and vitamin C, compared with control group B. After a follow-up of 1 year, there was a significant decline in total risk factors (32.0%) including hypertension, in the treatment arm compared with control subjects. Complications, such as obesity, hyperglycemia, hypercholesterolemia and electrocardiographic changes, were significantly lower in the intervention group compared to baseline and compared to control group B. It is clear that a prudent diet causes a significant decline in BPs and alteration in other risk factors leading to a decline in complications and decreased risk of CVDs and diabetes [34].

In another randomized trial, 217 hypertensive patients were administered either a 1600 Kcal/day diet rich in fruits, vegetables and legumes (group A, n = 108) or the usual 2100 Kcal/day diet (group B, n = 109) [35]. A follow-up, after 16 weeks, revealed that patients in the experimental group had significantly lower energy causing a 2.8 kg net decrease in mean body weight in conjunction with a significant net decline in mean BPs (7.5/6.5 mm Hg) compared with a non-significant reduction in the control group B. A significant net reduction in blood lipids; mean low-density lipoprotein (LDL)-cholesterol (7.9% mean total cholesterol (7.0%), and triglycerides (8.0%), was also noted, with a significant net rise in high-density lipoprotein (HDL)-cholesterol (4.0%) in the treatment group A compared with group B. Glucose intolerance (8.0%) and central obesity (waist-hip girth ratio, 0.021) also revealed a significant net decline in this group.

Epidemiological studies showed that an unhealthy diet rich in refined carbohydrates, salt, sugar, trans fat and saturated fat is also a risk factor for obesity and hypertension in South India [36], North India [37], as well as in other Asian countries [38]. Singh et al. also demonstrated that guava fruit which is rich in potassium, vitamin C, fiber and flavonoids, could cause a significant decline in blood pressure and blood lipoproteins as well as in oxidative stress [39]. These studies allowed us to develop diet and lifestyle guidelines for the prevention of hypertension among Indian on behalf of the Indian Society of Hypertension [40], which was also proven in an experiment on rabbits [41]. These data and other numerous studies published in the 21 century prompted other collaborators to highlight the cardio-protective effects of diet on CVDs and diabetes [42]. In later years, Singh and co-workers got an opportunity to propose that potassium and magnesium deficiency can aggravate blood pressures, leading to resistant hypertension [43] and a book on functional foods and nutraceuticals by Elsevier, Cambridge, MA, USA [44].

## 5. Effects of Japanese Diet on Hypertension

The Japan public health center-based prospective study, including 81,320 subjects aged 45–74 (36,737 men and 44,983 women), participated in a second survey in 1995–1998 and followed up till December 2012 [45]. Greater consumption of a diet rich in fruits, vegetables, soy products, potatoes, seaweed, mushrooms, and fish, was significantly associated with decreased risk of all-cause and cardiovascular disease mortality. The hazard ratios (95% confidence intervals) of all-cause and cardiovascular disease mortality for the highest versus lowest quartile of the prudent dietary pattern score were significant; 0.82 and 0.72, respectively. The intake of dairy products was also inversely associated with the risk of all-cause, cancer, and CVD mortality. A traditional Japanese dietary pattern was not associated with these risks. In another study involving 11,365 participants aged 20–84, three patterns of dietary intake, were identified: meat and fat, western type and traditional Japanese patterns [46]. The pattern of the traditional Japanese diet was closely associated with greater waist circumference, and BMI in males and greater diastolic BP in females. The pattern of the western type of diet was associated with lesser systolic BP but greater total and LDL cholesterol in both genders. The intake of dietary patterns with meat and fat was correlated with greater waist circumference, BMI, BP and blood lipids in both genders (trend *p* < 0.001).

A prospective cohort study involving 3486 men aged 30–71 in Osaka, Japan, was examined between 2001 and 2011 [47]. A follow-up after 4.6 years revealed that there were 846 new cases of hypertension. Those people eating meat less frequently had a greater risk of high BP (OR = 1.26) compared to participants who eat meat frequently. Reduced intake of dairy products on a daily basis also revealed a higher risk of high BP (OR = 1.39) compared with those who did eat daily. More frequent intake of dairy products and meat showed a decline in hypertension among males. In another cohort study comprised of 85,293 subjects, aged 45–74, followed up from 1998 to 2010 (1,305,738 person-years), 4110 strokes and 1283 cases of coronary artery disease occurred [48]. The authors observed no decreased risks of incident stroke in men or women with higher dietary magnesium intakes, but higher dietary magnesium intake was associated with a reduced risk of coronary artery disease in Japanese men. Thus, some of the epidemiological studies did not find an inverse association of fruit and vegetable intake with the risk of hypertension and stroke. Randomized, controlled trials are necessary to examine the role of fruits and vegetables or DASH diet on hypertension among Japanese.

Since the Japanese have a liking for a particular tastes and flavors, the adherence to the American DASH diet may be poor. In a 6-month study period, the 2-month treatments included the Japanese (J) J-DASH-1 diet 1×/day or the J-DASH-2 diet providing a fish hamburger patty 2×/day (5 days/ week respectively) [11]. The control group ate their usual diets, and in the later 4 months, all participants consumed their usual diets. In total, 40 people were randomized to the J-DASH 1 and J-DASH 2 groups (n = 13 each) and the control diet group (n = 14). Interestingly, clinic BP levels were not significantly different in the three groups, but the home BP values were lower in the J-DASH 1 group and lowest in the J-DASH 2 group, compared to the control-diet group and differed significantly among the three groups throughout the study period (*p* < 0.0001). The home BP variability was significantly lower in the J-DASH groups compared to the control-diet group throughout the study period (*p* < 0.01). It is clear that the J-DASH diet was suitable for reducing BP at home and decreasing its variations in the treatment group more effectively than the control group. The American Heart Association recommends the DASH diet to lower BP; however, its effects on Japanese participants have not been rigorously studied [12]. The effects of the DASH-Japan Ube Modified diet Program (DASH-JUMP), a modified DASH diet, were examined on cardiometabolic and inflammatory biomarkers in Japanese participants with untreated high-normal BP or stage 1 hypertension [12]. There were 58 patients, mean age 54.1 ± 8.1 years, with untreated high-normal BP or stage 1 hypertension, who followed the DASH-JUMP (salt 8.0 g per day) for 2 months. After the intervention period, they resumed their usual diets for 4 months. The DASH-JUMP significantly decreased the participants’ body mass index values from 24.6 ± 3.5 kg  m^−2^ to 23.2 ± 3.3 kg m^−2^ at 2 months, *p* = 0.000). BP, mean (153 ± 14/91 ± 11 mm Hg) also revealed a significant decline: 130 ± 16/80 ± 9 mm Hg at 2 months, *p* = 0.000 and 139 ± 16/85 ± 10 mm Hg at 6 months, *p* = 0.000). Fasting serum glucose and fasting insulin level also showed a significant decrease, indicating that the DASH-JUMP diet may be an effective nutritional strategy for preventing hypertension and cardio-metabolic risk.

## 6. Description of the Eating Plan of Various Types of Diets (Servings)

In a recent review, we compared the qualities of the DASH diet with a Mediterranean type of diet, an Indo-Mediterranean style diet and a Japanese diet, showing that latter two diets may be superior to the DASH diet [25]. The eating plan in the form of servings of functional foods also indicates that the Indo-Mediterranean diet may be superior to other diets due to the increased content of flavonoids in the diet. The Japanese diet eating plan also indicates its superiority due to the greater content of fish omega-3, fatty acids and fish peptides and tofu, which is rich in isoflavone. It seems that the DASH diet is a component of the Mediterranean diet, which has chosen certain foods rich in potassium with a lower intake of sodium. The quality of foods may vary in these diets along with their antihypertensive effects [39,45,46,47,48,49]. Table 2.

During the DASH eating plan, it is crucial to select foods that are: low in trans-fat and saturated fat as well as sodium & Rich in K, Ca, Mg, fiber, and protein. The study of the qualities of these four diets based on **twelve** high qualities of the best diet [25] also indicates that these characteristics are significantly higher in the Indo-Mediterranean style diet and Japanese diet compared to the other two diets [39,45,46,47,48,49]. Table 3.

## 7. Functional Foods and Nutrients with Antihypertensive Activity

Apart from drug therapy, healthy diet and exercise, self -care and treatment of emotional factors are effective in managing high BPs as well as BP variability [50,51,52,53]. The majority of the national guidelines include the DASH type of diet, a low-fat and low-sodium diet that encourages increased intake of fruits and vegetables. These diets are effective in controlling high BP, but adherence to the diet may be poor in various populations; hence there are few applicable dietary alternatives based on traditional foods rich in nutrients [53]. This is an important issue that can arise from poor health literacy in certain populations with a high risk of hypertension. There is an unmet need to outline the effect of specific dietary components, food and nutrients [54,55,56,57,58,59]—both positive and negative—when formulating a dietary approach for the management of high BPs that ultimately aims to improve patient adherence to the treatment and achieve better control of high BPs. Apart from the four types of diets, there are some individual fruits, such as guava and berries or vegetables, such as spinach, that may have independent BP-lowering effects [39,49,54,55,56,57,58,59,60,61]. Greater intake of K, Mg, carotene, polyunsaturated fatty acids, arginine and taurine, as well as vitamin D, along with exercise, may reduce BPs. Less-conclusive studies suggest that amino acids, tea, green coffee bean extract, dark chocolate, and foods high in nitrates may also reduce BP. The diet quality and its antihypertensive effect depend on the quantity of certain nutrients and foods that may be a component in all four diets.

## 8. Effects of Nutrients on Hypertension

In a meta-analysis, among 3960 subjects, out of 41 populations [54], cardiovascular risks were proven to be highly significantly lower in subjects who were excreting both 24 urine, taurine and magnesium, more than the averages despite differences in ethnicity and genetic background. It seems that taurine and magnesium are biomarkers for the intake of fish, other seafood, vegetables, soy, pulses, nuts, milk, etc., and increased dietary intake of these food sources could be recommended for the prevention of CVDs, including hypertension. In a single-arm clinical trial among 101 women, daily intake of lemon (vitamin C and potassium) combined with walking were effective for decreasing high BP because both showed a significant negative correlation to systolic BP Δ% [55]. Multiple linear regression analysis revealed that lemon ingestion is greatly associated with the blood citric acid level Δ% and the number of steps with BP Δ%, and it was surmised that the number of steps and lemon ingestion is related to improvement in BP.

In a cross-sectional study, among 2036 adults aged 25–41, the median index of omega-3 fatty acids was 4.58% [56]. The subjects in the highest omega-3 quintile had a systolic BP and diastolic BP that was 4 and 2 mmHg lower, compared with subjects in the lowest quartile of the omega-3 index, respectively (*p* < 0.01). There was an inverse linear relationship of the index of omega-3 fatty acids, with 24-h and office BP, which was significant. It is possible that a greater index of omega-3 may be associated with significantly lower BPs levels. It seems that diets rich in omega-3 fatty acids may be advised for reducing high BPs [56]. In a clinical study including 619 subjects aged from 40 years, the rate of high BP (140/90 mm Hg) and low serum 25(OH)D levels (<20 ng/mL) were respectively 55% and 32% [57]. The intake of calcium via diet was inversely correlated with hypertension in subjects with serum 25(OH) vitamin D levels higher than 20 ng/mL (OR: 0.995), but it was not significant in those with serum 25(OH)D levels of 20 ng/mL or lower. Intake of dietary vitamin D intake correlated with serum levels of 25(OH)D after adjustments for various confounding factors. It is possible that the regular intake of calcium may contribute to the prevention of hypertension in subjects with a non-vitamin D deficiency [57]. It seems that dietary vitamin D intake may effectively prevent this deficiency.

In a clinical trial, 125 patients (82 men and 43 women) with untreated mild or borderline hypertension were randomized to daily treatment with one of the following four regimens: 60 mmol K and 25 mmol (1000 mg) calcium, 60 mmol K and 15 mmol (360 mg) Mg, calcium and magnesium, or placebo [58]. At the entry to study, mean systolic and diastolic BPs were 139 ± 12 and 90 ± 4 mm Hg, respectively, and dietary intakes of K, calcium and Mg were 77 ± 32, 19 ± 13, and 12 ± 52 mmol/d, respectively. After treatment, the mean differences (with 95% confidence intervals) of the changes in BPs between the treatment and placebo groups were not significant, indicating only a modest effect of combinations of cation supplements in the treatment of BPs.

In another trial, 27 healthy subjects were administered either a high-nitrate (spinach; 845 mg nitrate/day) or low-nitrate soup (asparagus; 0.6 mg nitrate/day) for seven days with a 1-week washout period [59]. Treatment with high- vs. low-nitrate for seven days was associated with a decline in central systolic (−3.39 ± 5.6 mmHg, *p* = 0.004) and diastolic BP (−2.60 ± 5.8 mmHg, *p*= 0.028) and brachial systolic BP (−3.48 ± 7.4 mmHg, *p* = 0.022) at 180 min after therapy for seven days. It is possible that nitrate from spinach may provide beneficial hemodynamic effects targeted to the dietary approach to stop hypertension. The discovery of the nitrate–nitrite–nitric oxide (NO) pathway indicates that dietary (inorganic) nitrate has important vascular protective effects [60]. Nitrate from diets has been observed to reduce BP, inhibit aggregation of platelets, improve endothelial dysfunction and enhance exercise performance. Nitrite in the diet can protect against ischemia-reperfusion injury and decrease stiffness in the arteries, inflammation and intimal thickness [60]. There is evidence that nitrate in the diet may reduce the risk of CVDs, with diets high in nitrate-rich vegetables, such as the DASH diet, a Mediterranean diet, an Indo-Mediterranean diet, as well as the Japanese diet. It seems that there may be interactions with other nutrients, polyphenols, vitamin C and omega-3 fatty acids, which may enhance or inhibit these effects [60]. Nitrate intake may be improved by developing a table of vegetables showing nitrate content in various vegetables. Further large-scale studies are urgently needed to demonstrate the role of dietary nitrate in decreasing BP.

In a meta-analysis, seven meta-analyses were used for the umbrella review that reported significant positive benefits via treatment with l-arginine reducing systolic and diastolic BPs in hypertensive adults by 2.2 to 5.4 mm Hg and 2.7 to 3.1 mm Hg, respectively [61]. In pregnant women, diastolic BP with gestational hypertension also reduced by 4.9 mm Hg, apart from reducing the length of stay in the hospital for surgical patients. In addition, two of the three meta-analyses showed a 40% reduction in the incidence of hospital-acquired infections. These positive results should be considered with caution because statistically significant heterogeneity was observed in five of the seven meta-analyses. It is possible that treatment with l-arginine therapy for reducing systolic and diastolic BP in hypertensive may be useful. There is evidence that lycopene may improve vascular function and contributes to the primary and secondary prevention of CVDs [62]. The main activity profile of lycopene includes; antioxidant, anti-atherosclerotic, antiplatelet, anti-apoptotic, anti-inflammatory, antihypertensive and protective endothelial effects, the ability to improve the metabolic profile, and reduce arterial stiffness.

## 9. Effects of Foods on Hypertension

There are several traditional foods available in every country that appear to have antihypertensive effects [63,64,65,66,67,68,69,70]. In a clinical trial, 23 subjects were randomly assigned to either a pumpkin seed oil (PSO) (n = 12) or a placebo group (n = 11) [63]. Those patients in the PSO group consumed 3 g/day of PSO. The augmentation index, brachial and central systolic BP significantly (*p* < 0.05) decreased following PSO but not after placebo. Arterial stiffness and parameters of heart rate variability remained unchanged after PSO or placebo. It is clear that PSO can improve arterial hemodynamics in postmenopausal women and, therefore, might be effective in the prevention and treatment of hypertension.

In a meta-analysis, a total of 11 studies (14 trials) using flax seed were included [64]. The results revealed that flaxseed supplementation reduced systolic BP (−1.77 mm Hg; 95% CI: −3.45, −0.09 mm Hg; *p* = 0.04) and diastolic BP (−1.58 mm Hg; 95% CI: −2.64, −0.52 mm Hg; *p* = 0.003). These results were not influenced by the categorization of participants into higher baseline blood pressure (≥130 mm Hg). There was a significant improvement in diastolic BP in subgroup analysis for consuming whole flaxseed (−1.93 mm Hg; 95% CI: −3.65, −0.21 mm Hg; *p* < 0.05) and duration of consumption ≥ 12 weeks (−2.17 mm Hg; 95% CI: −3.44, −0.89 mm Hg; *p* < 0.05) [64]. It is clear that the intake of flaxseed can cause a slight decline in BP. The beneficial potential of flaxseed to reduce BP, in particular diastolic BP, may be greater when it is consumed as a whole seed and for a duration of >12 weeks.

In a crossover trial, 30 adults aged 40 to 74 with type 2 diabetes were included [65]. After a run-in period of 14 days, all patients took a low-fat diet (27% fat) having low-fat/high-carbohydrate snack foods and another group a moderate-fat diet (33% fat) having pistachios (20% of total energy) for 4 weeks, with a washout period of 2 weeks [65]. The group taking the pistachio diet, revealed significantly reduced total peripheral resistance (−3.7 ± 2.9%, *p* = 0.004), increased cardiac output (3.1 ± 2.3%, *p* = 0.002), and improved heart rate variability (all *p* < 0.05). Systolic ambulatory BP was significantly reduced by 3.5 ± 2.2 mm Hg (*p* = 0.046) following the pistachio diet, with the greatest reduction observed during sleep (−5.7 ± 2.6 mm Hg, *p* = 0.052). It is possible that a moderate-fat diet containing pistachios (MUFA) may cause a marginal decline in systolic BPs [65].

In an experimental study, taking carrot juice showed no beneficial effects. However, carrot juice intake reduced (*p* = 0.06) systolic BP, **without** any influence on diastolic BP. There was a significant (*p* < 0.05) increase in the plasma total antioxidant capacity and a decrease (*p* < 0.05) in the plasma generation of malondialdehyde [66]. Berries, such as Rosaceae (strawberry, raspberry, blackberry) and Ericaceae (blueberry, cranberry), have also been found to decrease BP [67]. The concerned bioactive ingredients in berries are mainly phenolics such as phenolic acids, flavonoids such as anthocyanins and flavonols, and tannins, as well as ascorbic acid. These compounds, either individually or combined, are responsible for various health benefits of various fruits and vegetables in the prevention of disorders of inflammation, CVDs, and chronic diseases [66,67,68,69,70,71,72,73,74,75,76].

## 10. Effects of Guava Fruits on Hypertension

Guava fruit, an important component of the Indo-Mediterranean diet, is included among the super-fruits, with a low-calorie profile and high dietary fiber, rich in antioxidant vitamins, flavonoids, vitamin C, potassium, lutein, beta carotene as well as minerals and vitamins [39,43,44]. Guava fruit is ingested with skin and is thought of as an apple for the poor in the tropics and sub-tropics. Each 100 g of guava fruits provides energy, 68 k calories, fiber 5.2 g, carbohydrates 11.2 g, protein 0.9, fat 0.3, calcium 10, phosphorous 28, iron 0.27, vitamin C 212 mg, riboflavin 0.040 mg, niacin 1.040 mg, vitamin B6 0.110 mg, folate 59 µg, sodium 5.5, magnesium 24 mg, potassium 91 mg, copper 0.14 mg, Zn 0.16 mg. Guava is a rich source of flavonoids (myricetin in sungkai fruit), 80.38 mg per 100 g dry weight of guava with the highest 93.75 mg in the skin of guava [44]. A high quantity of antioxidants was reported in maroon flesh cultivars with high total antioxidant activity (11.33 ±3.46 µmol), total flavonoids (82.69 ± 44.44 mg), total phenolics (168.21 ± 49.47 mg), and catechin (41.51 ± 14.98 mg) per 100 g [44].

Kumari et al. examined 45 students and randomly allocated 15 students in each of the 3 groups who received varying amounts of guava fruit. Of 45 subjects, 15 were supplemented with 400 g of ripe guava with peel (Group A) and 15 without peel (Group B) for 6 weeks [77]. The control group included the remaining 15 subjects. Treatment with guava fruit with peel revealed a significant (*p* < 0.05) reduction in systolic and diastolic BPs (*p* < 0.05) in group A, while Group B subjects also revealed a significant decrease (*p* < 0.05) in body mass index as well as BP. Singh et al. administered guava fruit among 61 group A and 59 group B patients with hypertension, preferably before meals in a foods-to-eat approach rather than a foods-to-restrict [39]. After a follow-up for 12 weeks, there was a significant net decline of 9.9% in serum total cholesterol, 7.7% in triglycerides (7.7%), as well as in systolic and diastolic BPs (9.0/8.0 mm Hg). There was a significant 8.0% net increase in high-density lipoprotein cholesterol after 12 weeks of guava fruit therapy. (Table 4).

Table 5 indicates that higher consumption of guava fruit may also result in a significant rise in serum K and Mg with a significant decrease in serum sodium [39]. It is clear that a part of the reduction in BPs may be on account of these nutrients. (Table 5)

## 11. Effects of Food Groups; Fruits, Vegetables and Pulses

Fruits and vegetables are major components of the DASH diet as well as the Indo-Mediterranean style diet. A meta-analysis included eight equal energy intake trials (n = 554 patients with and without hypertension), with significant heterogeneity for all outcomes [68]. Interestingly, pulses in the diet, exchanged for other foods, with the same calories, significantly reduced systolic (−2.25 mm Hg, *p* = 0.03) and mean arterial BP (−0.75 mm Hg, *p* = 0.03) and diastolic BP non-significantly (−0.71 mm Hg, *p* = 0.17). In the USA, the objective of the study was to determine the cellular antioxidant activity in 25 commonly consumed fruits [69]. The contents of total phenolic and oxygen radical absorbance capacity (ORAC) values of berries (wild blueberry, blackberry, raspberry, and blueberry) and pomegranate showed the highest value of antioxidant activity, whereas the lowest values were observed for banana and melons. Interestingly, the largest contributors of fruit phenolics were apples, and the biggest suppliers of cellular antioxidant activity in the diet were apples and strawberries [69]. It is clear that an increased intake of fruits may be healthy advice to enhance the consumption of flavonoids to reduce oxidative stress, which may cause a decline in the risk of CVDs and cancer. In another meta-analysis [70], 28 reports were included, showing inverse association for 30 g whole grains/d (RR: 0.92), 100 g fruits/d (RR: 0.97), 28 g nuts/d (RR: 0.70) and 200 g dairy/d (RR: 0.95) with the risk of hypertension. A positive association was also observed for 100 g red meat/d (RR: 1.14), 50 g processed meat/d (RR: 1.12), and 250 mL sugar sweetened beverages (SSBs)/d (RR: 1.07) [70]. This study showed that optimal consumption of fruits, whole grains, nuts, legumes, dairy, red and processed meats, and SSBs related to the risk of high BPs. It is clear that among commonly consumed foods, raw vegetables, tomatoes, carrots, and scallions are related significantly and inversely to BP. However, among commonly eaten cooked vegetables, tomatoes, peas, celery, and scallions related significantly **and** inversely to BP [71]. It seems that antioxidant activity should be measured using biologically relevant assays that are important in the screening of fruits and vegetables for health benefits [70].

## 12. Effects of Flavonoids on Hypertension

It is possible that flavonoid content of the various diets could be an important determinant of decline in BPs. Most of the flavonoids, as isolated from outside of the food matrix, from the five main subgroups consumed (flavones, flavonols, flavanones, flavan-3-ols, and anthocyanins, may have antihypertensive effects along with their other effects [72,73,74], Flavonoids from all five subgroups have been shown to attenuate a rise in or to reduce BP in hypertension, diabetes, metabolic syndrome, and stroke. The potential mechanisms for regulating BP are found in most flavonoids, in particular flavones, flavonols, flavanones, anthocyanins and flavanols, which were able to modulate BP by restoring endothelial function, either directly by affecting nitric oxide levels or indirectly, through other pathways [73,74,75]. Cohort studies indicate an inverse association between flavanone intake and risk of ischemic stroke, flavonol consumption and risk of diabetes and anthocyanin intake and risk of myocardial infarction [75]. Randomized controlled trials among humans show that catechins and quercetin impart significant beneficial effects on BPs. The majority of the flavonoids, in particular cocoa flavonoids, mediate their antihypertensive effects by increasing the bioavailability of nitric oxide, reducing endothelial cell oxidative stress or modulating the activity of vascular ion channels [73,74,75]. Interestingly, out of nine studies, only three found significant changes, with one out of two studies on subjects with hypertension and two of the six on subjects with prehypertension [76]. All other remaining trials were made in subjects without high BP and showed no decrease in BP.

In a meta-analysis, the antihypertensive effects of a common flavonoid, quercetin, were examined in seven trials comprised of 587 patients [78]. The findings showed a significant decline in both the systolic BP (3.04 mm Hg, *p* = 0.028) and diastolic BP (2.63 mm Hg, *p* < 0.001) following supplementation with quercetin. On sub-categorization according to the quercetin dose, there was a significant systolic BP and diastolic BP-reducing effect in randomized controlled trials with doses ≥500 mg/day (4.45 mm Hg, *p* = 0.007 and 2.98 mm Hg, *p* < 0.001, respectively). There was a lack of a significant effect for doses <500 mg/day, but indirect comparison tests failed to significant differences between doses. It seems that quercetin had the most consistent BP-lowering effect in animal and human studies, irrespective of diseases, dose and duration. Further research on the safety and efficacy of the flavonoids is required before any of them can be used by humans, presumably in supplement form, at the doses required for therapeutic benefit. There is some research indicating that many spices, and herbs that are rich in flavonoids, can reduce BPs in subjects with hypertension or prehypertension [79]. The exact intake of flavonoids in various countries is not known. The content of flavonoids appears to be significantly greater in the Indo-Mediterranean diet compared to the DASH diet and other healthy diets because per day flavonoid intake among subjects eating the Indo-Mediterranean diet was 1185 mg/day, which is much higher compared to Mediterranean countries (95 mg/day), USA (200–250 mg/day) and Japan (87 mg/day) indicating, high quality of this diet.

## 13. Mechanisms on the Role of Diets Influencing Blood Pressures

It seems that the elasticity of vessel walls, peripheral vascular resistance, cardiac output, the volume of blood in circulation and viscosity of blood are important physio-pathological factors that determine the level of BPs. Since electrolytes; sodium, potassium, magnesium and calcium have a potential influence on smooth muscle cell contraction, a reduction in sodium intake with an increased intake of potassium and magnesium is considered useful in the prevention and treatment of high BP [1,2,3]. Hypertension is associated with the interplay of many factors, including environmental, genetic, adaptive, anatomic, neural, endocrine, humoral, and hemodynamic mechanisms [52]. Oxidative stress is characterized by excessive production of reactive oxygen species (ROS) and an altered oxidation-reduction (redox) state, leading to a marked increase in inflammation [80]. An increase in inflammation, in conjunction with oxidation of proteins and dysregulation in cell signaling, may lead to a further increase in inflammation, followed by proliferation and apoptosis, leading to migration and fibrosis. These molecular alterations are important processes contributing to impaired vascular function, cardiovascular remodeling, renal dysfunction, immune cell activation, and sympathetic nervous system excitation in the development of hypertension [80]. The increase in free radical stress may cause injury in the endothelium and dysregulate endothelium-dependent relaxation in the vessels, causing a rise in vascular contraction and predisposing hypertension.

It is possible that free radical stress is a common biochemical mechanism in hypertension, because other factors, such as angiotensin II (Ang II), endothelin-1 (ET-1), aldosterone (Aldo), western diet and salt (Na) induce activation of NADPH oxidases (Noxs) which are known to increase ROS. These oxidants may influence multiple systems, including microbiota, involved in the pathophysiology of hypertension [52,79,80,81,82,83]. Western style diets are known to predispose free radical stress by elevating the levels of carbonylation of protein with the increased level of lipid peroxides while decreasing the status of antioxidant defense [81]. Free radical stress is produced by an imbalance in the redox state of the cell, either by overproduction of ROS or by dysfunction of the antioxidant systems, dependent on diet and lifestyle factors (Figure 2).

Apart from western diet-induced oxidative stress, increased salt intake in conjunction with other risk factors of hypertension can also cause an increase in brain RAAS, which increases oxidative stress with increased angiotensin II signaling in the neuronal cells, in the brain. There is increased activity of the sympathetic nervous system leading to elevation in BPs. Intervention with an Indo-Mediterranean diet decreases oxidative stress with a decline in sympathetic activity with a reduction in BPs, as given in Figure 3.

In addition, the gut microbiota may communicate with the nervous, endocrine, and immunity systems for the regulation of host homeostasis, including kidney functions and BPs [79,84]. There is a connection between various organs in the brain–gut–kidney axis that are mediated by descending autonomic regulation from the brain and signals from the gut and the kidney, such as immune products and microbial metabolites. Thus, the gut–kidney axis is connected via metabolism-dependent and immune pathways, which regulate BP and kidney function [79]. Figure 3.

Further evidence shows that heightened sympathetic nervous system (SNS) activity, especially in the kidney and brain, increases BP in obese patients [84]. Adipokines, including adiponectin and leptin, and the renin–angiotensin–aldosterone system (RAAS)may also contribute to hypertension. High salt diet-induced adiponectin may decrease sodium/glucose cotransporter (SGLT) 2 expression in the kidney, which causes a decrease in BP. A high salt diet may also alter the release of adipokines and RAAS-related components, linking the gastrointestinal tract to BP. Glucagon-like peptide-1 (GLP-1) and ghrelin decrease BP in both rodents and humans [84]. The sweet taste receptor in enteroendocrine cells increases SGLT1 expression and stimulates sodium/glucose absorption [84]. Intestinal mineralocorticoid receptors also regulate sodium absorption and BP; however, gastric sensing of sodium induces natriuresis, and gastric distension increases BP. A western type of diet, high in salt, sugar and fat, may disrupt the gut barrier, causing systemic inflammation, a decline in GLP-1 and leading to insulin resistance, with increased BP.

It seems that gut microbiota regulates BP by secreting vasoactive hormones and short-chain fatty acids on increased intake of foods rich in fibers and flavonoids, which are similar to molecules that are released on the administration of probiotics and prebiotics. Bariatric surgery also improves metabolic disorders and hypertension due to increasing in GLP-1 secretion, decreasing leptin secretion and SNS activity, and changing gut microbiome composition [84]. Other therapies targeting the gastrointestinal system may be an Indo-Mediterranean type diet for improving metabolic abnormalities and reducing BP. It is possible that SNS, the brain, adipocytes, RAAS, angiotensin II, the kidney, the gastrointestinal tract, and microbiota play important roles in regulating BP. Therefore, the gut microbiota could be a novel target for the treatment of hypertension as a metabolic disorder via novel therapeutic strategies, including dietary interventions, probiotics and prebiotics [84].

## 14. Discussion

There is evidence that hypertension is a disease that predisposes heart attacks, heart failure, stroke and chronic kidney disease, which are the biggest killers [1,2,3]. It seems that two-thirds of premature deaths due to chronic diseases are connected with four shared modifiable behavioral risk factors: tobacco intake, western type diet, lower physical activity and alcoholism, which are also risk factors for hypertension. However, the content of this review is diet specific. Apart from genetic and environmental factors such as pollution, these unhealthy behaviors predispose key metabolic changes that increase the risk of hypertension, obesity, diabetes, and other CVDs. Most of the premature deaths from these diseases are largely preventable by enabling health systems to respond more effectively and equitably to the health care needs of people. It is crucial to have public health policies influencing all behavioral risk factors, in particular, a healthy diet that can inhibit **salt toxicity**, glucotoxicity and lipotoxicity, at an affordable cost, to the population for the prevention of immature deaths due to hypertension and its complications. It seems that all the traditional diets may be healthy, but the addition of salt, sugar, refined foods, and red and preserved meat makes these diets unhealthy. The Japanese diets or other healthy diets may only be popular in Japan and less popular in other Asian countries. Hence traditional foods of each country, rich in antihypertensive nutrients and foods, should be substituted for unhealthy foods. Guava fruit is rich in nutrients, in particular flavonoids, fiber, **potassium** and vitamin C, which have beneficial effects on BPs and other risk factors. Guava fruit is not a component of the DASH diet and other healthy diets, but it may be included in a higher quantity similar to the Indo-Mediterranean diet.

There is evidence that Indo-Mediterranean foods such as berries, guava and apple may have a hepato-protective effect against oxidants; NAFLD, in part by decline or increase in the oxidant/antioxidant balance with a decrease in pro-inflammatory markers concerned with the progression of the disease through a decline in the generation of ROS [82,83]. Recently, treatment with α-linolenic acid and other omega-3 fatty acids (rich in the brain) that may elevate the sensitivity of insulin with increased vagal activity as well as inhibit adipogenesis have been found to inhibit oxidative stress and inflammation and cause a decrease in BPs [83]. In view of these findings, it is imperative to include fruit, vegetables, whole grains, millets, soya beans, barley, porridge, spices, fish and nuts in the guidelines for the prevention of hypertension and other CVDs [79,84]. It is proposed that these healthy diets can also decrease western diet-induced oxidative stress and inflammation in the brain, causing a decline in the brain renin–angiotensin–aldosterone system (RAAS), and angiotensin II, leading to an increase in vagal nerve activity and a decline in sympathetic nerve activity with a decrease in BPs.

In an earlier meta-analysis involving 24 trials comprised of 23 858 participants with high BP, the DASH diet found the greatest net decrease in systolic BP, −7.62 mm Hg and diastolic BP, −4.22 mm Hg [85]. The effects of low-sodium; low-sodium, high-potassium; low-sodium, low-calorie; and low-calorie diets also revealed a significant decline in systolic and diastolic BP. Surprisingly, subjects in the Mediterranean diet group experienced a significant incremental reduction in only diastolic but not systolic BP. It seems that this meta-analysis gives the message that the pattern of the DASH diet was quite effective for the regulation of BPs, which was as good as drug therapy by a single drug [85]. Interestingly, the Mediterranean diet, which is known to reduce the rate of cardiovascular events and mortality, showed only little effect on the control of BP. However, it is possible that a longer duration of the study and greater systolic BP at baseline may reveal an increased decline in BP due to a Mediterranean style diet [28,29]. It is also likely that taking more antihypertensive Indo-Mediterranean flavonoid-rich foods, such as guava, blackberries, millets, herbs and spices, in particular, cinnamon, turmeric, fenugreek and oregano, may cause lower BPs because these foods can decrease BPs readings within 24 h [25,74,75,76,77,86].

Although the DASH type of diet has been demonstrated to cause a significant decline in total and cause-specific mortality [87,88], the Mediterranean diet using the DASH score also showed a significant decline in total and cause-specific mortality [3,89]. Interestingly, the bias for the DASH diet has been so strong that a meta-analysis, including 17 studies, also found that even modest adherence to the DASH diet was associated with a lower risk of all-cause and cause-specific mortality [88]. However, a pattern of diet identical to two dietary patterns, the DASH diet and the Mediterranean diet (DASH score > median and aMED > median), was related to a reduced risk of death among patients with diabetes [3]. In this study, a total of 28,905 participants were analyzed, and 2598 of them died after a median follow-up of 6.3 years. Surprisingly, adherence to the Mediterranean diet (aMED >3 vs. ≤3), but not the DASH diet, was associated with a significant decline in the risk of all-cause mortality (adjusted HR 0.74, 95% CI 0.66–0.83, *p* < 0.001) in the overall population [3]. Diet could be the treatment of choice in patients with pre-hypertension diagnosed with lower limits of BPs as per new guideline by AHA [90]. The Japanese-Brazilian Diabetes Study Group found a significant association between peripheral arterial disease, metabolic syndrome and homocysteine with high total fat intake, low intake of fiber from fruit and oleic acid, independently of other variables, indicating that the Japanese diet may be superior to other diets [91,92,93].

## 15. Limitations

It seems that the experimental data provided in the review are incomparable because the results of experiments are from different test groups, the age ranges of the patients are different, the experimental methods and the criteria for judging hypertension are different, and the years of the experiments are also different. Ideally, the experimental observation of different diets should be carried out on the same subject group, and all experimental methods, judgment standards and scoring standards should be consistent. Finally, the proportion of hypertensive patients can be compared to directly reflect which diet has a better prevention effect on hypertension.

## 16. Conclusions

An unhealthy diet, in particular, high salt, sugar and saturated fat intake, without an optimal vegetable and fruit intake, physical inactivity, alcoholism, tobacco, sleep and emotional disorders are major risk factors for hypertension. Apart from an excess of sodium in the diet, unhealthy western type diets appear to have independent adverse effects on vascular function by causing oxidative stress and inflammation **leading to an impaired vascular function** and an increase in BPs. Oxidative stress due to glucotoxicity and lipotoxicity is characterized by excessive production of ROS and an altered oxidation-reduction (redox) state, leading to a marked increase in inflammation in the brain, gut and vessels. The DASH diet, established for the prevention and treatment of hypertension, does not appear to be superior to the Mediterranean diet in reducing BPs and CVDs. The Mediterranean type of diet and Indo-Mediterranean diet, apart from providing potassium and calcium, may also have increased additional beneficial effects due to greater fiber and flavonoids on gut microbiota, which is known to regulate BPs. Since the Indo-Mediterranean diet and Mediterranean diet contain more flavonoids and fiber, and the Japanese diet is rich in fish omega-3 fatty acids and fish peptides, it poses the possibility that these diets are superior to the DASH diet in reducing BPs, as well as CVDs and all-cause mortality. Larger randomized, controlled trials and cohort studies would be necessary to confirm our views.

## Figures and Tables

**Figure 1 nutrients-15-00046-f001:**
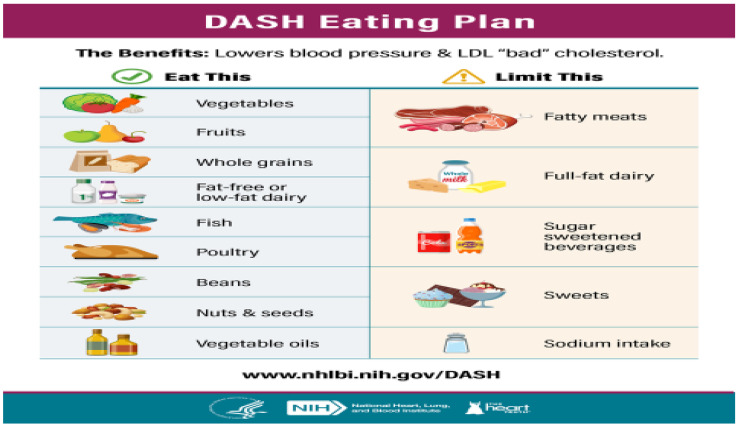
Eating plan in the DASH diet showing foods that should be consumed either in excess or under limits (Adapted from [13], www.nhlbi.nih-gov/DASH).

**Figure 2 nutrients-15-00046-f002:**
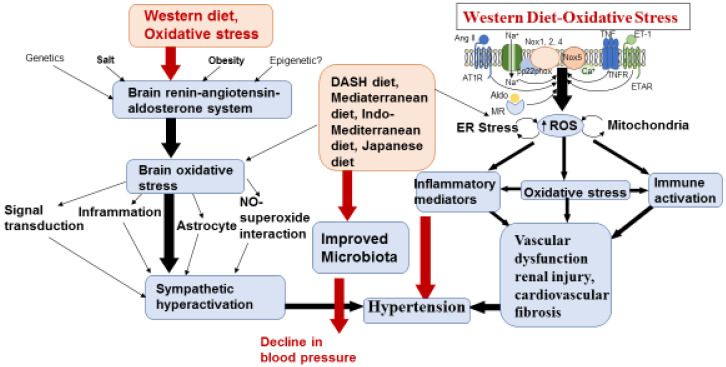
**Western diet induced** oxidative stress and inflammation causing increase in brain renin-angiotensin-aldosteron system (RAAS) predisposing vascular dysfunctions and hypertension.

**Figure 3 nutrients-15-00046-f003:**
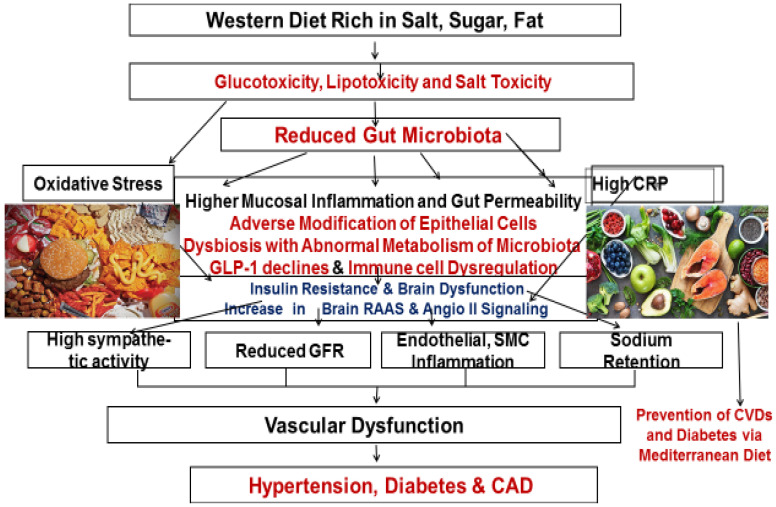
Western diet reduces gut microbiota and dysregulate gut-brain-kidney-axis in the development of hypertension.

**Table 1 nutrients-15-00046-t001:** Effects of Indo- Mediterranean foods; fruits, vegetables and legumes on blood pressures in patients receiving thiazide diuretics. (Modified from Singh et al. [32]).

Data	Intervention Group (n = 97)	After 1 Year	Control Group (n = 100)	After 1 Year
Blood Pressure, mm Hg				
Systolic	152.2 ± 12.8	148.2 ± 10.1	154.6 ± 9.8	160 ± 12
Diastolic	99.8 ± 7.2	90.2 ± 4.8	97.0 ± 4.8	103 ± 5.6
Serum magnesium, mg/dL	1.52 ± 0.24	1.86 ± 9.2	1.54 ± 0.23	1.6 ± 0.11
Serum potassium, mEq/L	4.1 ± 0.42	4.86 ± 0.45	4.5 ± 0.43	4.00 ± 0.44
Resistant hypertension, n (%)	97	5 (5.19)	100	17 (17.9)

**Table 2 nutrients-15-00046-t002:** Description of the DASH type Eating Plan (Servings) compared with other diets.

	DASH Diet	Indo-Mediterranean	Mediterranean Diet	Japanese Diet
Whole grains	6–8	5–6	5–6	4
Vegetables	4–5	4–5	4–5	4–5
Fruits	4–5	4–5	4–5	4–5
Meats & poultry	6 or less	Not advised	Low	Low
Fish	1–2 weekly	2–3, weekly	moderate	High, 3–4
Fats and oils	2–3	3–4	3–4	3–4
Low-fat or fat-free dairy products.	2–3	2–3	2–3	Not advised
Sodium	2300 mg	>2300, reduce	>2300, Reduce	>2300, Reduce
Sweets	5 or less weekly	Not advised or Jaggery, honey	Not advised	Not advised
Nuts, seeds, dry beans and peas.	4–5, weekly	7–8	5–6	Not advised
Nutrients	High K, no emphasis on flavonoids	High K & flavonoids	High K & flavonoids	High K, flavonoids Omega-3, fish peptides

1500 milligrams (mg) of sodium lowers blood pressure even further than 2300 mg sodium daily.

**Table 3 nutrients-15-00046-t003:** The **twelve** qualities of the other high-quality diets, compared to the DASH diet.

Quality of Foods	DASH Diet	Indo-Mediterranean Diet	Mediterranean Diet	Japanese Diet
Low glycemic index	Very low	Low	Low	Very low
High nutrient density	High	Very high	High	Very high
Food diversity	High	Very high	High	High
No trans fat	mild	No	mild	No
No/low sugar or refined	mild	No	mild	No
Low salt	Low	Low	Low	Low
Moderate healthy fat	Vegetable oils	Rape seed oil	Olive oil	Rice bran oil
High fiber, flavonoids	Moderate	High	Moderate	Moderate
Good for gut microbiota	Good	Very good	Good	Good
No peroxidation of foods	Mild	No	Mild	No
No red/preserved meat	Low	No	Low	No
Foods, need mastication	Moderate	Heavy	Moderate	Moderate
Additional qualities	None	Spices, millets, porridge	None	Fish, vegetables, whole rice

**Table 4 nutrients-15-00046-t004:** Effects of guava fruit intake on blood glucose, blood pressure and blood lipids and estimated 10 year risk of coronary disease among patients with hypertension. Risk score is based on the Framingham score.

Data	Intervention Group A (n = 61)		Control Group B (n = 59)	
	Baseline	After 12 Weeks	Baseline	After 12 Weeks
Fasting blood glucose, mg/dL	108 ± 10	−10.7 *	110 ± 11	−5.2
Total cholesterol, mg/dL	226 ± 20	−27.6 *	223 ± 18	−5.6
Triglycerides, mg/dL	166 ± 18	−18.6 *	162 ± 15	−5.8
High-density lipoprotein cholesterol, mg/dL	44 ± 8	+2.4 *	46 ± 11	−1.2
Blood pressure, mm Hg				
Systolic	163 ± 6	−13.6 *	161 ± 7	−4.6
Diastolic	106 ± 4	−11 *	104 ± 5	−3.0
Body weight, Kg	67 ± 9	−2.2	69 ± 11	−1.8
Estimated 10 years risk of CAD,%				
Male	52 ± 19	−13.4 *	50 ± 22	−2.6
Female	30 ± 12	−10.3 *	36 ± 15	−2.1

* = Significant * = *p* < 0.01, by two sample *t*-test using analysis of variance. Modified from reference [30].

**Table 5 nutrients-15-00046-t005:** Effects of guava fruit intake on macro-minerals related to blood pressures.

Data	Intervention Group (n = 61)		Control Group (n = 59)	
	Base Line	After 12 Weeks	Baseline	After 12 Weeks
Sodium, mEq/L	144 ± 12	−6.8 *	149 ± 14	−4.5
Potassium, mEq/L	4.2 ± 0.4	+0.6 *	4.5 ± 4	−0.2
Calcium, mg/dL	9 ± o.5	+0.06	9 ± 0.5	−0.1
Serum magnesium, mg/dL	1.65 ± 0.3	+0.06 *	1.6 ± 0.3	+0.5
Serum albumin, mg/dL	4.4 ± 0.5	−0.1	4.4 ± 0.5	+0.2

* = *p* value < 0.05, significant by two sample *t*-test and analysis of variance by comparing changes in the two groups (modified from reference [39]).

## Data Availability

Data support statement not required as it is a review article.

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
