# Peer review of "Dietary Approaches to Stop Hypertension via Indo-Mediterranean Foods, May Be Superior to DASH Diet Intervention"

_nutrients, 2022, doi:10.3390/nu15010046_

Round 1

Reviewer 1 Report

Comments: The logic of the full text is chaotic, the hierarchy of the article is not clear, and the paragraph titles cannot reflect the logic and hierarchy of the full text. The titles of the first part of the article are: Effects of Mediterranean Type of Diet in Hypertension, Effects of Indo-Mediterranean Style Diet in Hypertension, Effects of Japanese Diet in Hypertension. The title of the previous article reflects that the author wants to describe the effect of different diets on hypertension. The following titles are Description of the Eating Plan of Various Types of Diets; Functional foods and nutrients with Antihypertensive activity; Effects of nutrients; Effects of food; Effects of Guava fruits; Effects of Food Groups; fruits, vegetables and pulses; Effects of Flavonoids. The latter title is seriously disconnected from the former title, and cannot be connected, the logic of the title is chaotic, and the level is not clear. The reviewer could not understand why the roles of Guava fruits; fruits, vegetables and pulses; and Flavonoids were described in detail separately. According to personal understanding, these common nutritional active substances should be included in different diets, such as flavonoids, phenolics, vitamins, etc. , just in different proportions. However, the authors did not compare and describe the proportions of these beneficial active substances in different diets at all. Furthermore, the author's description may lead readers to misunderstand that these beneficial active substances may only be present in the Indo-Mediterranean diet. The author needs to rewrite and logically organize the full text so that readers can understand it better. In addition, the whole review is full of grammatical errors and the formats are not uniform. Thus, this review need to rewrite carefully.

(1) The format of full-text references is not uniform. Eg, refs. 83, 84, 85, 90, 91, 92, the name of the journal is in italics, as well as in normal, and the document format needs to be unified. In addition, reference 86 (line 980-984), the interval between reference is strange.

(2) The Figures in the article are not clear, and some Figures are out of proportion, such as Figure 1, which needs to be re-adjusted.

(3) The grammatical errors are common in the full article, such as line 30-32(the abstract part), there is a logical problem. From the reviewer's point of view this sentence means that sedentary behavior, and western diet with high salt, sugar, alcoholism, and smoking can protect against cardiovascular disease, including hypertension. The description is inconsistent with the logic, please revise it. Besides, there are a lot of other grammatical errors in the full text. Eg, Line 774, should be the Indo-Mediterranean diet, not the Indp-Mediterranean diet, please check the full text carefully. The font format of line 610-618 is different, please modify it.

(4) Abstract section, line 36-37, the author mentioned the causes of hypertension induced by Western type foods, what foods Indo-Mediterranean foods and Japanese foods contain, and the beneficial effects of Japanese foods on the prevention of hypertension (Line 47-52). The entire abstract part does not mention at all what is the main reason why Indo-Mediterraean Food is better than DASH Diet in terms of the intervention of hypertension. The abstract fails to reflect the central theme of the article and does not correspond to the title of the article. In addition, what foods are included in Indo-Mediterranean foods and Japanese foods may be more appropriate being included in the Introduction section. The abstract section should mainly describe the important findings and viewpoints of the review.

(5) Abstract section, Line 38-39, the authors stated that the main reasons for the different proportions of hypertensive patients in different countries are diet quality, lack of functional foods, and completely ignoring differences in population genes and environments. The authors need to emphasize in the Discussion section that the content of this review is primarily diet-specific to avoid misunderstandings among readers. In addition, the authors need to discuss in the review what the main causes of hypertension are, and the proportion of dietary, environmental, or genetic factors that contribute to the induction of hypertension. If the effect of different diets on the induction of hypertension is not the most obvious, then this part also needs to be mentioned in the discussion section to help readers better understand the content.

(6) Keywords cannot reflect the key content of the article. According to the title of the article, full text should emphasize the improvement effect of Indo-Mediterranean diet on hypertension, but the keywords wrote the Mediterranean diet. Please clarify what is the relationship between Indo-Mediterranean diet and the Mediterranean diet, are they same, and if not, what are the differences?

(7) The main theme of the full text is not unified, and the content of the full text does not fit well with the title of the article. Based on the title of the article, the review should be focus on the role of the Indo-Mediterranean diets on hypertension and what are the salient advantages of the diet compared to other diets. Although the authors described the effects of different diets on hypertension separately, they do not conduct a detailed and systematic comparison of the advantages and disadvantages of these four diets (DASH diet, Indo-Mediterranean diets, Mediterranean diets, and Japanese diet). In addition, the experimental data provided in the review are incomparable, the results of experiments are from different test groups, the age ranges of the patients are different, the experimental methods and the criteria for judging hypertension are different, and the years of the experiments are also different. The data are scattered. And lack of intuitive data, no intuitive graphs or tables were provided to compare the different diets and to emphasize the superior advantages of Indo-Mediterranean diets. The experimental observation of different diets should be carried out on the same subject group. And all experimental methods, judgment standards and scoring standards should be consistent. Finally, the proportion of hypertensive patients can be compared to directly reflect which diet has a better prevention effect on hypertension.

(8) Conclusion section, Line 774-776, the author mentioned that Japanese diet, Indo-Mediterranean diet and Mediterranean diet are better than DASH diet in improving hypertension, but the title of the article only mentions that Indo-Mediterranean diet is better than DASH diet , the conclusion of the review does not correspond well with the title of the article.

(9) The full-text data is limited. The probability of hypertensive patients is only mentioned in the United States, the Mediterranean region, and Japan, without the data from other Asian countries, or European countries. The data cannot fully reflect the beneficial effects of Indo-Mediterranean diets on hypertension, and its superiority over the DASH diet. In addition, Japanese diets may only be popular in Japan and less popular in other Asian countries. Therefore, the authors need to provide information on the acceptance of these diets in different countries and the proportion of the corresponding hypertensive population in order to better reflect the effect of diets on prevention of hypertension.

(10) Introduction section, line 59-63, the author mentions that the criteria for judging hypertension in the United States and some Mediterranean regions are (systolic pressure of >= 140 mmHg and or diastolic pressure >= 90 mm Hg), and according to this criterion, Since 1999, the probability of hypertensive patients has remained at about 30%. Subsequently, due to the low new standard of hypertension (ref. 3, Line 63), the probability has risen to 47%. However, the reviewer noticed that the ref.3 was published in 2018 and was aimed at the American population. Therefore, the reviewer wondered whether the hypertension judgment criteria in the United States have been updated in recent years, and whether the criteria are universal in the world, and at the same time what is the specific criterion for this judgment, the author should provide this information to help readers make better understanding and judgment.

(11) Introduction section, lines 63-65, the author should provide the prevalence of hypertension in the Mediterranean region to help readers better compare the effects of different diets on hypertension.

(12) Introduction section, line 87-90, the description in this section fails to truly reflect the gist of the review. The author mentioned that the main purpose of the review is to emphasize the role of DASH diet in hypertension, and other healthy diets in other countries have potential anti-inflammatory and antihypertensive activity. This section does not mention Indo-Mediterranean foods which was included in the title of the article. Therefore, the author should carefully confirm what the subject of this article is, and to write according to the subject of the article.

(13) Line 522-548, the author described in detail the nutrient composition ratio of Guava fruit, mentioning its effect on hypertensive patients. I was wondering if this food exists only in the Indo-Mediterranean diet, and whether other diets do not contain this food. In addition, according to the author's description, Guava fruit may be beneficial to the human body because it contains high content of antioxidants, such as phenolics, flavonoids, and catechin. The author should provide the amounts and proportions of these nutrients in other diets style. Also, the DASH diet may not contain Guava fruit, but it may include other foods that provide a lot of antioxidants. Therefore, the author also should provide this part of the content for the convenience of readers' understanding. In addition, line 535 shows that it is an observational experiment conducted on students, and the experimental data in this part does not seem to reflect the beneficial effect of the Indo-Mediterranean diet on hypertension. Other experimental observation data are almost all from the elderly group of 50-70 years old. The physical health status of different age groups is different. I don't understand what this part of the data means in the review.

Author Response

Comments: The logic of the full text is chaotic, the hierarchy of the article is not clear, and the paragraph titles cannot reflect the logic and hierarchy of the full text. The titles of the first part of the article are Effects of Mediterranean Type of Diet in Hypertension, Effects of Indo-Mediterranean Style Diet in Hypertension, Effects of Japanese Diet in Hypertension. The title of the previous article reflects that the author wants to describe the effect of different diets on hypertension. The following titles are Description of the Eating Plan of Various Types of Diets; Functional foods and nutrients with Antihypertensive activity; Effects of nutrients; Effects of food; Effects of Guava fruits; Effects of Food Groups; fruits, vegetables and pulses; Effects of Flavonoids. The latter title is seriously disconnected from the former title, and cannot be connected, the logic of the title is chaotic, and the level is not clear. The reviewer could not understand why the roles of Guava fruits; fruits, vegetables and pulses; and Flavonoids were described in detail separately. According to personal understanding, these common nutritional active substances should be included in different diets, such as flavonoids, phenolics, vitamins, etc. , just in different proportions. However, the authors did not compare and describe the proportions of these beneficial active substances in different diets at all. Furthermore, the author's description may lead readers to misunderstand that these beneficial active substances may only be present in the Indo-Mediterranean diet. The author needs to rewrite and logically organize the full text so that readers can understand it better. In addition, the whole review is full of grammatical errors and the formats are not uniform. Thus, this review need to rewrite carefully. Tried to improve.

(1) The format of full-text references is not uniform. Eg, refs. 83, 84, 85, 90, 91, 92, the name of the journal is in italics, as well as in normal, and the document format needs to be unified. In addition, reference 86 (line 980-984), the interval between reference is strange. Corrected

(2) The Figures in the article are not clear, and some Figures are out of proportion, such as Figure 1, which needs to be re-adjusted. Figure legend explained

(3) The grammatical errors are common in the full article, such as line 30-32(the abstract part), there is a logical problem. From the reviewer's point of view this sentence means that sedentary behavior, and western diet with high salt, sugar, alcoholism, and smoking can protect against cardiovascular disease, including hypertension. I do not agree because its action plan of WHO to implement: Line 1, but I am changing. The WHO action plan is to alter; sedentary behavior, western type diet with high salt and sugar, tobacco and alcoholism, for prevention of cardiovascular diseases (CVDs) including hypertension.

The description is inconsistent with the logic, please revise it. Besides, there are a lot of other grammatical errors in the full text. Eg, Line 774, should be the Indo-Mediterranean diet, not the Indp-Mediterranean diet, please check the full text carefully. The font format of line 610-618 is different, please modify it. Managing Editor, please help.

(4) Abstract section, line 36-37, the author mentioned the causes of hypertension induced by Western type foods, what foods Indo-Mediterranean foods and Japanese foods contain, and the beneficial effects of Japanese foods on the prevention of hypertension (Line 47-52). The entire abstract part does not mention at all what is the main reason why Indo-Mediterraean Food is better than DASH Diet in terms of the intervention of hypertension. Sir, its given in lines, 12-17

The abstract fails to reflect the central theme of the article and does not correspond to the title of the article. In addition, what foods are included in Indo-Mediterranean foods and Japanese foods may be more appropriate being included in the Introduction section. The abstract section should mainly describe the important findings and viewpoints of the review. We have included concerned foods in lines 12-17 as well as findings and views.lines, 21-23

(5) Abstract section, Line 38-39, the authors stated that the main reasons for the different proportions of hypertensive patients in different countries are diet quality, lack of functional foods, and completely ignoring differences in population genes and environments. Its added in the abs. line 9

The authors need to emphasize in the Discussion section that the content of this review is primarily diet-specific to avoid misunderstandings among readers. In addition, the authors need to discuss in the review what the main causes of hypertension are, and the proportion of dietary, environmental, or genetic factors that contribute to the induction of hypertension. If the effect of different diets on the induction of hypertension is not the most obvious, then this part also needs to be mentioned in the discussion section to help readers better understand the content. Given pn page 19, line 710-713

(6) Keywords cannot reflect the key content of the article. According to the title of the article, full text should emphasize the improvement effect of Indo-Mediterranean diet on hypertension, but the keywords wrote the Mediterranean diet. Please clarify what is the relationship between Indo-Mediterranean diet and the Mediterranean diet, are they same, and if not, what are the differences? Differences given in the abstract, line 12-17, Indo-Mediterranean diet is already given in title, no point to repeat in key words.

(7) The main theme of the full text is not unified, and the content of the full text does not fit well with the title of the article. Based on the title of the article, the review should be focus on the role of the Indo-Mediterranean diets on hypertension and what are the salient advantages of the diet compared to other diets. Although the authors described the effects of different diets on hypertension separately, they do not conduct a detailed and systematic comparison of the advantages and disadvantages of these four diets (DASH diet, Indo-Mediterranean diets, Mediterranean diets, and Japanese diet). Sir, This is a selected review not a meta-analysis, advantages of each diet are given in tables 2 and 3 as well as in separate subtitles, pages 5-9. Disadvantages of DASH diet given Page 3, line 120 showing low preserved meat and sugar also indicated in table 2 and 3 for all the 4 diets.

In addition, the experimental data provided in the review are incomparable, the results of experiments are from different test groups, the age ranges of the patients are different, the experimental methods and the criteria for judging hypertension are different, and the years of the experiments are also different. The data are scattered. This point is discussed under Limitations at the end page 19.

And lack of intuitive data, no intuitive graphs or tables were provided to compare the different diets and to emphasize the superior advantages of Indo-Mediterranean diets. The experimental observation of different diets should be carried out on the same subject group. And all experimental methods, judgment standards and scoring standards should be consistent. Finally, the proportion of hypertensive patients can be compared to directly reflect which diet has a better prevention effect on hypertension. Given under limitations. Tables 2 and 3 are given comparing different types of diets, titles corrected.

(8) Conclusion section, Line 774-776, the author mentioned that Japanese diet, Indo-Mediterranean diet and Mediterranean diet are better than DASH diet in improving hypertension, but the title of the article only mentions that Indo-Mediterranean diet is better than DASH diet , the conclusion of the review does not correspond well with the title of the article. The title, would look much larger if Japanese diet is also given, please read the Introduction and elsewhere but can add it if necessary. Conclusion modified, all 4 diets may be equally good

(9) The full-text data is limited. The probability of hypertensive patients is only mentioned in the United States, the Mediterranean region, and Japan, without the data from other Asian countries, or European countries. The data cannot fully reflect the beneficial effects of Indo-Mediterranean diets on hypertension, and its superiority over the DASH diet. In addition, Japanese diets may only be popular in Japan and less popular in other Asian countries. Therefore, the authors need to provide information on the acceptance of these diets in different countries and the proportion of the corresponding hypertensive population in order to better reflect the effect of diets on prevention of hypertension. The Japanese diets or other healthy diets may only be popular in Japan and less popular in other Asian countries, hence traditional foods of each country, rich in antihypertensive nutrients and foods should be substituted for unhealthy foods, given in the discussion, para 1.

(10) Introduction section, line 59-63, the author mentions that the criteria for judging hypertension in the United States and some Mediterranean regions are (systolic pressure of >= 140 mmHg and or diastolic pressure >= 90 mm Hg), and according to this criterion, Since 1999, the probability of hypertensive patients has remained at about 30%. Subsequently, due to the low new standard of hypertension (ref. 3, Line 63), the probability has risen to 47%. However, the reviewer noticed that the ref.3 was published in 2018 and was aimed at the American population. Therefore, the reviewer wondered whether the hypertension judgment criteria in the United States have been updated in recent years, and whether the criteria are universal in the world, and at the same time what is the specific criterion for this judgment, the author should provide this information to help readers make better understanding and judgment. Sentence deleted to remove confusion.

(11) Introduction section, lines 63-65, the author should provide the prevalence of hypertension in the Mediterranean region to help readers better compare the effects of different diets on hypertension. Prevalence of hypertension is same in Mediterranean countries (30%) as in USA.

(12) Introduction section, line 87-90, the description in this section fails to truly reflect the gist of the review. The author mentioned that the main purpose of the review is to emphasize the role of DASH diet in hypertension, and other healthy diets in other countries have potential anti-inflammatory and antihypertensive activity. This section does not mention Indo-Mediterranean foods which was included in the title of the article. Therefore, the author should carefully confirm what the subject of this article is, and to write according to the subject of the article. Revised, This selected review aims to highlight the role of DASH diets, as compared with Indo-Mediterranean, Japanese and Mediterranean type of diets, in the management of hypertension, because certain foods and nutrients may have potential anti-inflammatory and antihypertensive activity as determinant of diet quality.

(13) Line 522-548, the author described in detail the nutrient composition ratio of Guava fruit, mentioning its effect on hypertensive patients. I was wondering if this food exists only in the Indo-Mediterranean diet, and whether other diets do not contain this food. In addition, according to the author's description, Guava fruit may be beneficial to the human body because it contains high content of antioxidants, such as phenolics, flavonoids, and catechin. The author should provide the amounts and proportions of these nutrients in other diets style. Also, the DASH diet may not contain Guava fruit, but it may include other foods that provide a lot of antioxidants. Therefore, the author also should provide this part of the content for the convenience of readers' understanding. Flavonoids mentioned in table  3,Guava fruit is rich in nutrients, in particular flavonoids, fiber and vitamin C which have beneficial effects on BPs and other risk factors. Such fruits may also be component of DASH diet and other healthy diets but in lower quantity compared to Indo-Mediterranean diet, given under discussion para 1.

In addition, line 535 shows that it is an observational experiment conducted on students, and the experimental data in this part does not seem to reflect the beneficial effect of the Indo-Mediterranean diet on hypertension. Other experimental observation data are almost all from the elderly group of 50-70 years old. The physical health status of different age groups is different. I don't understand what this part of the data means in the review. Given in the limitations. This part of data is given to emphasize that these foods can also reduce BPs and hence are included in the Indo-Mediterranean diet.

Added under Flavonoid Intake, last para. The exact intake of flavonoids in various countries are not known. The content of flavonoids appears to be significantly greater in the Indo-Mediterranean diet compared to DASH diet and other healthy diets, because per day flavonoid intake among subjects eating Indo-Mediterranean diet was 1185 mg/day, which is much higher compared to Mediterranean countries (95 mg/day), USA (200-250 mg/day) and Japan (87 mg/day) indicating, high quality of this diet.

Submission Date

08 November 2022

Date of this review

17 Nov 2022 09:02:37

Reviewer 2 Report

1 – Review abstract. The first paragraph does not bring either the reason for the review or the objective.

2 – Line 58: Add which Mediterranean countries the authors refer to.

3 – Lines 73 to 75: The goals of “The WHO Global NCD action plan is a road map with policy options to 73 to be implemented from 2013 to 2020 focusing on the 4 shared, modifiable behavioral risk 74 factors that are linked to the 4 preventable NCDs” were achieved?

4 – Line 57 to 90: Review English language in all text.

5 – Review the title of the work. The objective is to carry out a review. The title of the work leads the reader to believe that it is an empirical study.

6 – Huge paragraphs making it difficult to read. To correct.

7 – The work does not have the item “method”. Therefore, there is a lack of information about the type of review (scope, systematic, meta-analysis), whether such a review was submitted to formal registration, the period of the review, which databases were searched, which keywords were used, which outcome was used, which Boolean operators are adopted.

8 – Add description of the original DASH diet and adapted to other countries, to contextualize comparisons.

9 – Add a summary description of the studies presented (where, when, sample, instruments and main results).

10 – Add algorithms (article collection flow).

11 – Add information (chart or table with the results of data collection and the scores obtained for each article and their bias).

12 – Correct the P to p-value or p throughout the work.

13 – Line 105: What are vegetables? Could it be legumes? To correct.

14 – The figures are out of focus making reading impossible.

15 – Correct punctuation throughout the text.

16 – Lines 312 to 327: See, for example, that the authors speak of the prospective cohort study and compare it to another cohort study (without informing whether the study is prospective or retrospective), without informing the reader where such study was carried out or even if they are comparable. Why do the authors find it surprising that meat and dairy consumption is associated with a decline in hypertension? Was this association significant?

17 – Lines 312 to 327: The studies presented compare different nutrients. Does not make sense.

18 – I suggest reviewing the study presentation strategy. There is no way to analyze the differences between diets if there is no description of them. In the case of the Mediterranean diet there is a generic description on line 105, but it is not enough.

19 – Check studies that compare the diet of the Japanese-Brazilian population, for example:

Garofolo, L., Barros Jr, N., Miranda Jr, F., D'Almeida, V., Cardien, L.C., & Ferreira, S.R. (2007). Association of increased levels of homocysteine ​​and peripheral arterial disease in a Japanese-Brazilian population. European journal of vascular and endovascular surgery, 34(1), 23-28.

Gimeno, S.G.A., Hirai, A.T., Harima, H.A., Kikuchi, M.Y., Simony, R.F., de Barros Jr, N., ... & Japanese-Brazilian Diabetes Study Group. (2008). Fat and fiber consumption are associated with peripheral arterial disease in a cross-sectional study of a Japanese-Brazilian population. Circulation Journal, 72(1), 44-50.

Damiao, R., Castro, T.G., Cardoso, M.A., Gimeno, S.G., Ferreira, S.R., & Japanese–Brazilian Diabetes Study Group. (2006). Dietary intakes associated with metabolic syndrome in a cohort of Japanese ancestry. British journal of nutrition, 96(3), 532-538.

20 – Lines 356 to 358: “In a recent review, we have compared the qualities of DASH diet with Mediterranean type of diet, Indo-Mediterranean style diet and Japanese diet, showing later two diets may be superior to DASH diet [25”] . We have compared??? Is the article non-review? “showing later two diets may be superior to DASH diet” which two? Review essay. Self-citation is not appropriate.

21 – Lines 715 to 718: Review writing. Confused paragraph.

22 – Discussions are superficial and do not deserve a review article. In addition, after presenting so many studies, it was expected that, in the discussion, the authors would make it clear whether the analyzed diets are or are not promoters of hypertension reduction.

23 - In general, despite an extensive review, the authors need to inform the methodological issues pointed out, the figures, tables and suggested tables and an adequate structure of internal division that gives the reader a better understanding of the text.

Author Response

1 – Review abstract. The first paragraph does not bring either the reason for the review or the objective. Sir,its important to give background but modified; Western type diet with high salt and sugar, sedentary behavior, obesity, tobacco and alcoholism, are important risk factors of hypertension.

2 – Line 58: Add which Mediterranean countries the authors refer to. Greece and Spain

3 – Lines 73 to 75: The goals of “The WHO Global NCD action plan is a road map with policy options to 73 to be implemented from 2013 to 2020 focusing on the 4 shared, modifiable behavioral risk 74 factors that are linked to the 4 preventable NCDs” were achieved?

4 – Line 57 to 90: Review English language in all text. Revised

5 – Review the title of the work. The objective is to carry out a review. The title of the work leads the reader to believe that it is an empirical study. Added selected review above the title.

6 – Huge paragraphs making it difficult to read. To correct. edited

7 – The work does not have the item “method”. Therefore, there is a lack of information about the type of review (scope, systematic, meta-analysis), whether such a review was submitted to formal registration, the period of the review, which databases were searched, which keywords were used, which outcome was used, which Boolean operators are adopted. Method is not needed because it is a selected review. No registration done. We used DASH diet, Japanese diet, Mediterranean diet, fruits, vegetables and nutrient rich diet in hypertension as key words for search, then selected the studies.

8 – Add description of the original DASH diet and adapted to other countries, to contextualize comparisons. Given in tables 2 and 3.also described on Page 3, para 2.

9 – Add a summary description of the studies presented (where, when, sample, instruments and main results).Thank you, done

10 – Add algorithms (article collection flow).Not required as it is a selected Review

11 – Add information (chart or table with the results of data collection and the scores obtained for each article and their bias). Its selected review hence such table not required.

12 – Correct the P to p-value or p throughout the work. Revised

13 – Line 105: What are vegetables? Could it be legumes? To correct. See page 5, para 1,fruits; guava, black berry, apples, vegetables; green leafy, gourds, onion, garlic and legumes; green beans, black beans, kidney beans along with mustard oil  in decreasing BPs

14 – The figures are out of focus making reading impossible. edited

15 – Correct punctuation throughout the text. Ok, thank you

16 – Lines 312 to 327: See, for example, that the authors speak of the prospective cohort study and compare it to another cohort study (without informing whether the study is prospective or retrospective), without informing the reader where such study was carried out or even if they are comparable. Why do the authors find it surprising that meat and dairy consumption is associated with a decline in hypertension? Was this association significant? Surprising Deleted, association was significant.

17 – Lines 312 to 327: The studies presented compare different nutrients. Does not make sense. We mentioned the nutrients that were assessed by the concerned authors.

18 – I suggest reviewing the study presentation strategy. There is no way to analyze the differences between diets if there is no description of them. In the case of the Mediterranean diet there is a generic description on line 105, but it is not enough. Greater description of DASH diet is given in Figure 1, Table 2 and table 3, also added on Page 3, para 2, The DASH diet is rich in fruits; berries, apple, oranges, bananas, appricots vegetables; green leaves, green bean whole grains; oats, beans and low-fat dairy foods, milk, yogurt, cheese and vegetable oils, as well as low red meat, egg and fish

19 – Check studies that compare the diet of the Japanese-Brazilian population, for example: These references included at the end and few lines about findings on page 19, last para

 Garofolo, L., Barros Jr, N., Miranda Jr, F., D'Almeida, V., Cardien, L.C., & Ferreira, S.R. (2007). Association of increased levels of homocysteine ​​and peripheral arterial disease in a Japanese-Brazilian population. European journal of vascular and endovascular surgery, 34(1), 23-28.

 Gimeno, S.G.A., Hirai, A.T., Harima, H.A., Kikuchi, M.Y., Simony, R.F., de Barros Jr, N., ... & Japanese-Brazilian Diabetes Study Group. (2008). Fat and fiber consumption are associated with peripheral arterial disease in a cross-sectional study of a Japanese-Brazilian population. Circulation Journal, 72(1), 44-50.

Damiao, R., Castro, T.G., Cardoso, M.A., Gimeno, S.G., Ferreira, S.R., & Japanese–Brazilian Diabetes Study Group. (2006). Dietary intakes associated with metabolic syndrome in a cohort of Japanese ancestry. British journal of nutrition, 96(3), 532-538.

20 – Lines 356 to 358: “In a recent review, we have compared the qualities of DASH diet with Mediterranean type of diet, Indo-Mediterranean style diet and Japanese diet, showing later two diets may be superior to DASH diet [25”] . We have compared??? Is the article non-review? “showing later two diets may be superior to DASH diet” which two? Review essay. Self-citation is not appropriate. We compared the composition of diets in terms of foods and nutrients and given our references only for Indo-Mediterranean diet because its our hypothesis, sorry about it.

21 – Lines 715 to 718: Review writing. Confused paragraph.Corrected: It seems that gut microbiota regulates BP by secreting vasoactive hormones and short-chain fatty acids on increased intake of foods rich in fibers and flavonoids which are similar to molecules that are released on administration of probiotics and prebiotics.

22 – Discussions are superficial and do not deserve a review article. In addition, after presenting so many studies, it was expected that, in the discussion, the authors would make it clear whether the analyzed diets are or are not promoters of hypertension reduction. Improved

23 - In general, despite an extensive review, the authors need to inform the methodological issues pointed out, the figures, tables and suggested tables and an adequate structure of internal division that gives the reader a better understanding of the text. We used DASH diet, Japanese diet, Mediterranean diet, fruits, vegetables and nutrient rich diet in hypertension as key words for search, then selected the studies.

Submission Date

08 November 2022

Date of this review

11 Nov 2022 17:34:2

Round 2

Reviewer 1 Report

ok

Reviewer 2 Report

Dear authors,

The text now presented is more precise and concise. I congratulate you on the effort to treat such important data at a time when the path towards sustainability and social justice is being discussed. After the corrections made, I am in favor of publishing the text.